# Spatial patterning of P granules by RNA-induced phase separation of the intrinsically-disordered protein MEG-3

Jarrett Smith, Deepika Calidas, Helen Schmidt, Tu Lu, Dominique Rasoloson, Geraldine Seydoux*

Department of Molecular Biology and Genetics, Howard Hughes Medical Institute, Johns Hopkins University School of Medicine, Baltimore, United States

**Abstract** RNA granules are non-membrane bound cellular compartments that contain RNA and RNA binding proteins. The molecular mechanisms that regulate the spatial distribution of RNA granules in cells are poorly understood. During polarization of the *C. elegans* zygote, germline RNA granules, called P granules, assemble preferentially in the posterior cytoplasm. We present evidence that P granule asymmetry depends on RNA-induced phase separation of the granule scaffold MEG-3. MEG-3 is an intrinsically disordered protein that binds and phase separates with RNA in vitro. In vivo, MEG-3 forms a posterior-rich concentration gradient that is anti-correlated with a gradient in the RNA-binding protein MEX-5. MEX-5 is necessary and sufficient to suppress MEG-3 granule formation in vivo, and suppresses RNA-induced MEG-3 phase separation in vitro. Our findings suggest that MEX-5 interferes with MEG-3's access to RNA, thus locally suppressing MEG-3 phase separation to drive P granule asymmetry. Regulated access to RNA, combined with RNA-induced phase separation of key scaffolding proteins, may be a general mechanism for controlling the formation of RNA granules in space and time.

**\*For correspondence:** gseydoux@jhmi.edu

**Competing interests:** The authors declare that no competing interests exist.

## Introduction

RNA granules are concentrated assemblies of RNA and RNA-binding proteins that form without a limiting membrane in the cytoplasm or nucleoplasm of cells (*Courchaine, 2016*). RNA granules are ubiquitous cellular structures and several classes of cytoplasmic RNA granules have been described, including stress granules, P bodies, neuronal granules and germ granules (*Anderson and Kedersha, 2006*). Cytoplasmic RNA granule components typically exchange rapidly between a highly concentrated pool in the granule and a more diffuse, less concentrated pool in the cytoplasm (*Weber and Brangwynne, 2012*). In addition to RNA-binding domains, proteins in RNA granules often contain prion-like, low complexity, or intrinsically-disordered regions (IDRs) (*Courchaine, 2016*). In concentrated solutions, IDRs spontaneously de-mix from the aqueous solvent to form liquid droplets (liquid-liquid phase separation or LLPS) or hydrogels (*Li et al., 2012*; *Weber and Brangwynne, 2012*; *Elbaum-Garfinkle et al., 2015*; *Guo and Shorter, 2015*; *Lin et al., 2015*; *Nott et al., 2015*). Like RNA granules in vivo, proteins in LLPS droplets and hydrogels exchange with the solvent (*Kato et al., 2012*; *Li et al., 2012*; *Elbaum-Garfinkle et al., 2015*; *Lin et al., 2015*). These findings have suggested that LLPS or reversible gelation drives the assembly of RNA granules in vivo (*Guo and Shorter, 2015*).

In cells, RNA granule assembly is regulated in space and time. For example, stress granules assemble within seconds of exposure to toxic stimulants that require the temporary removal of mRNAs from the translational pool (*Anderson and Kedersha, 2006*). In eggs, germ granules assemble in the germ plasm, a specialized area of the cytoplasm that is partitioned to the nascent germline

**eLife digest** Animal cells contain many smaller compartments known as organelles that perform particular roles. For example, a compartment called the nucleus stores most of the cell's genetic information. The nucleus and many other organelles form inside layers of membrane that physically separate them from the rest of the cell. However, some organelles, such as the germ granule, do not have a membrane. It is thought that these organelles may form in the same way that oil droplets tend to come together when mixed with water. However, oil droplets form in water spontaneously and do not fall apart, so it is not clear how cells could control the assembly and destruction of such organelles.

The germ granules inside the cells of a worm called *C. elegans* are destroyed and reassembled in cycles. Smith et al. investigated how the worm cells control these cycles. The experiments show that a protein called MEG-3 is required to allow the components of granules to transition from behaving like individual molecules dissolved in water (similar to being dissolved in cell fluid) to assembling into droplets. When MEG-3 is mixed with molecules of ribonucleic acid (RNA) it can bind very tightly to the RNA and then separate out from the rest of the fluid to form distinct droplets.

Smith et al. also show that another protein called MEX-5 can destroy these droplets by attaching itself to RNA in place of MEG-3, which causes MEG-3 to dissolve back into the rest of the fluid. The physical properties of the MEG-3 droplets are still not known and so the next step following on from this work will be to find out whether germ granules behave like liquids, gels or hard solids.

during the first embryonic cleavages (*Voronina et al., 2011*). How phase separation, a spontaneous process in vitro, is regulated in vivo to ensure that RNA granules form at the correct place and time is not well understood.

The germ (P) granules of *C. elegans* are an excellent model to study the mechanisms that regulate granule assembly (*Updike and Strome, 2010*). For most of development, P granules are stable perinuclear structures, but in the transition from oocyte-to-embryo, P granules detach from the nucleus and become highly dynamic (*Pitt et al., 2000*; *Wang et al., 2014*). As the oocyte is ovulated in the spermatheca, P granules disassemble and release their components in the cytoplasm. After fertilization, P granule proteins reassemble into dynamic granules that undergo repeated cycles of assembly and disassembly in synchrony with cell division. Live imaging in the 1-cell zygote has revealed that these cycles are spatially patterned along the anterior-posterior axis of the embryo: granule assembly is favored in the posterior and granule disassembly is favored in the anterior (*Brangwynne et al., 2009*; *Gallo et al., 2010*). By the first mitosis, P granules are found exclusively in the posterior cytoplasm together with other germ plasm components. As a result, P granules are segregated to the posterior germline blastomere $P_1$ and excluded from the anterior somatic blastomere AB.

P granule asymmetry is under the control of the PAR polarity network which divides the zygote into distinct anterior and posterior domains (*Motegi and Seydoux, 2013*). The PAR-1 kinase is enriched in the posterior cytoplasm and restricts the RNA-binding protein MEX-5 (and its redundant homolog MEX-6) to the anterior cytoplasm (*Griffin et al., 2011*). MEX-5 and MEX-6 in turn restrict P granules to the posterior (*Schubert et al., 2000*; *Gallo et al., 2010*). In *mex-5 mex-6* double mutants, P granule still undergo cycles of assembly and disassembly but these are no longer patterned along the anterior-posterior axis, and small granules remain throughout the cytoplasm (*Gallo et al., 2010*). An attractive hypothesis is that MEX-5 blocks phase separation of P granule components in the anterior cytoplasm (*Brangwynne et al., 2009*; *Lee et al., 2013*). The mechanism of MEX-5 action and the critical P granule component(s) regulated by MEX-5, however, are not known.

P granule assembly in zygotes requires several P granule proteins, including the RNA-binding protein PGL-1 (and its redundant paralog PGL-3) and the intrinsically-disordered protein MEG-3 (and its redundant paralog MEG-4) (*Hanazawa et al., 2011*; *Wang et al., 2014*). PGL-1 and PGL-3 are RGG domain proteins that self-associate and recruit other RNA-binding proteins to the granules, including the GLH family of RNA helicases (*Updike and Strome, 2010*; *Hanazawa et al., 2011*).

MEG-3 and MEG-4 are redundant, serine-rich proteins that bind to PGL-1 in vitro and are essential for P granule assembly in embryos. In zygotes lacking *meg-3* and *meg-4*, PGL-1 and GLH-2 form transient assemblies that do not segregate asymmetrically and are not maintained in later stages (*Wang et al., 2014*). Phosphorylation of MEG-3 by the DYRK kinase MBK-2 promotes granule disassembly in zygotes, but the mechanism that favors disassembly specifically in the anterior cytoplasm is not known (*Wang et al., 2014*). In this study, we use a combination of in vivo and in vitro experiments to examine the contribution of the MEX, MEG, and PGL proteins to P granule assembly and asymmetry. We show that MEG-3/4, but not PGL-1/3, are essential for granule assembly and asymmetry, and that MEX-5 localizes MEG-3 in a posterior-rich gradient. We demonstrate that MEG-3 is an RNA-binding protein that is stimulated by RNA to undergo phase separation. MEX-5 is sufficient to block MEG-3 phase separation in vitro and to block MEG-3 granule formation in vivo. Our findings are consistent with a model whereby MEX-5 antagonizes MEG-3's access to RNA to inhibit MEG-3 condensation in the anterior cytoplasm.

## Results

### Hierarchical regulation of P granule assembly

To determine the genetic hierarchy that controls granule asymmetry, we first compared the distributions of MEX-5, MEG-3, PGL-1 and GLH-1 during the earliest stages of zygote polarization. We monitored MEX-5, MEG-3 and GLH-1 localization using tagged alleles generated by genome editing (Materials and methods, *Supplementary file 1* ) and PGL-1 localization using a polyclonal antibody that recognizes PGL-1 (*Strome and Wood, 1983*).

Before polarization, MEX-5, MEG-3, PGL-1 and GLH-1 were all distributed evenly throughout the cytoplasm. MEX-5 appeared mostly diffuse in the cytoplasm with a few foci, whereas MEG-3, PGL-1, and GLH-1 appeared both diffuse and enriched in many small (<1 micron diameter) foci (*Figure 1A*). During polarization (pronuclear formation and migration), MEX-5 and MEG-3 began to redistribute into opposing cytoplasmic gradients along the long axis of the zygote (anterior-posterior axis) with MEG-3 beginning to form large (~1 micron) granules in the posterior (*Figure 1A*). Total levels of MEG-3 do not change during this period, consistent with redistribution of existing MEG-3 protein from anterior to posterior (*Figure 1B*, *Figure 1—figure supplement 1*). In contrast to MEX-5 and MEG-3, PGL-1 and GLH-1 remained uniformly distributed during polarization. After polarization (mitosis), all proteins were localized, with MEX-5 in the anterior cytoplasm and MEG-3, PGL-1 and GLH-1 in large granules in the posterior cytoplasm (*Figure 1A*).

To determine the interdependence of these localizations, we examined the effect of removing MEX-5/6, MEG-3/4 or PGL-1/3 using RNA-mediated interference (RNAi) or genetic mutants (GLH-1 has already been shown to depend on PGL-1/3 for asymmetry [*Hanazawa et al., 2011*]). In zygotes derived from mothers treated with double-stranded RNA against *mex-5* and *mex-6* (*mex-5/6(RNAi)* zygotes), the MEG-3 gradient did not form and MEG-3 and PGL-1 granules remained uniformly distributed throughout the cytoplasm (*Figure 1C*, (*Gallo et al., 2010*). In *meg-3; meg-4* double mutant embryos, the MEX-5 gradient was unaffected but neither PGL-1 nor PGL-3 granules segregated properly (*Wang et al., 2014*), *Figure 1C*, *Figure 1—figure supplement 1*). In *pgl-1(RNAi); pgl-3 (bn104)* zygotes, the MEG-3 and MEG-4 gradients were unaffected and MEG-3/MEG-4 granules formed in the posterior as in wild-type, except that the granules appeared smaller especially in zygotes (*Wang et al., 2014*), *Figure 1C*, *Figure 1—figure supplement 1*). These analyses suggest that MEX-5 and MEX-6 regulate granule asymmetry by localizing MEG-3 and MEG-4 to the posterior, which in turn are required to localize PGL-1 and PGL-3. PGL-1 and PGL-3 are not required to localize MEG-3 or MEG-4 or to assemble MEG-3/4 granules, but contribute to the size of MEG-3/4 granules as reported previously (*Wang et al., 2014*).

### MEX-5 is necessary and sufficient to suppress MEG-3 granule formation

Using *mex-5* transgenes, *Griffin et al. (2011)* showed that the formation of the MEX-5 gradient requires phosphorylation of serine 404 in the C-terminus of MEX-5 by the kinase PAR-1. To determine whether the MEX-5 gradient is required to pattern MEG-3, we mutated serine 404 to alanine (S404A) at the *mex-5* locus using CRISPR/Cas9-mediated genome editing (*Paix et al., 2015*). We introduced the S404A mutation in two strains: one where the *mex-5* locus had been previously

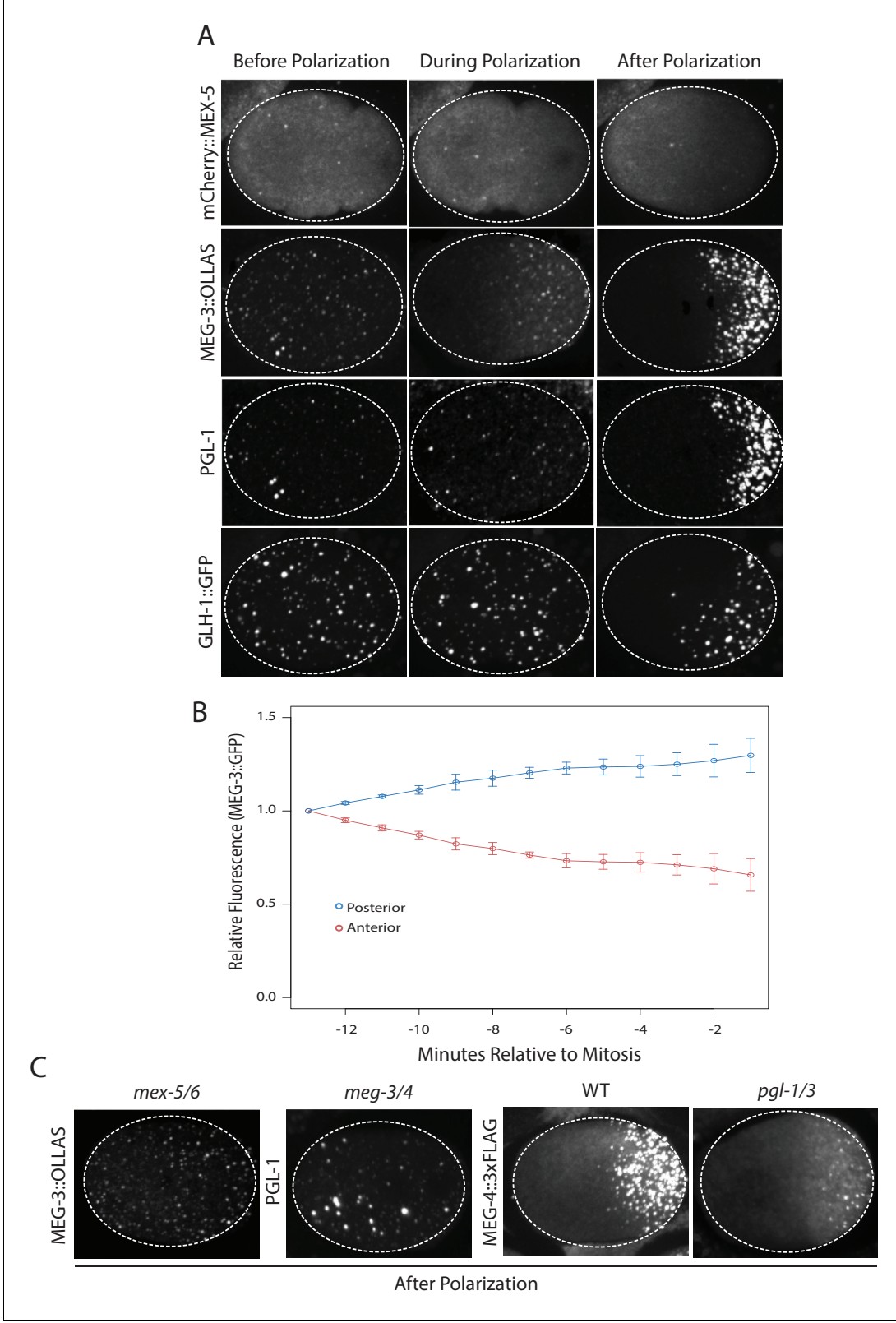

**Figure 1.** Localization of P granule proteins during zygote polarization. (**A**) Photomicrographs of live wild-type (mCherry::MEX-5 and GLH-1::eGFP) or fixed *meg-4* zygotes (MEG::3 OLLAS and PGL-1) at three different stages: before polarization (pronuclear formation), during polarization (pronuclear migration) and after polarization (mitosis). *meg-4(ax3052)* zygotes were co-immunostained for MEG-3::OLLAS (anti-OLLAS, Novus Biological), and PGL-1 (K76, DSHB). *meg-4* is required redundantly with *meg-3* for P granule assembly, and each is sufficient to support localized granule assembly

*Figure 1 continued on next page*

*Figure 1 continued*

(**Wang et al., 2014**). In this and subsequent figures, dashed lines outline each embryo, embryos are oriented with anterior to the left and posterior to the right and are ~50 µM long. At least three embryos were examined per genotype shown. (B) MEG-3::meGFP levels in the anterior and posterior halves of the zygote during polarization. Values represent average fluorescence intensity over time (relative to initial levels) in the anterior (red) and posterior (blue). Averages come from values measured from three different embryos. Error bars represent standard deviation of the mean. (C) Photomicrographs of fixed zygotes after polarization immunostained for OLLAS, PGL-1, or FLAG. *mex-5/6* zygotes were derived from wild-type hermaphrodites treated with *mex-5* and *mex-6* RNAi. *meg-3/4* zygotes were derived from *meg-3(ax3055); meg-4(ax3052)* hermaphrodites. *pgl-1/3* zygotes were derived from *pgl-3(bn104)* hermaphrodites treated with *pgl-1* RNAi (see **Figure 1—figure supplement 1** for additional examples of *pgl-1 (RNAi);pgl-3(bn104)* zygotes also stained for PGL-1 to verify loss of PGL-1).

The following figure supplement is available for figure 1:

**Figure supplement 1.** MEG-3 localizes before PGL-1 and does not require PGL-1 or PGL-3 to assemble granules.

tagged with mCherry to monitor MEX-5 localization, and one where MEG-3 had been previously tagged with GFP to monitor MEG-3 localization (**Supplementary file 1** ). As expected, we found that MEX-5(S404A) failed to form a gradient during zygote polarization and remained uniformly distributed (**Figure 2A**). Using the MEG-3::GFP strain, we found that zygotes derived from mothers homozygous for *mex-5(s404a)* (*mex-5(S404A)* zygotes), MEG-3 did not form a gradient or granules. Instead, MEG-3 remained uniformly distributed in the cytoplasm throughout the 1-cell stage (**Figure 2B**). We conclude that MEX-5 is sufficient to suppress the formation of MEG-3 granules throughout the cytoplasm.

## MEX-5 RNA binding is required to suppress MEG-3 granule assembly

The MEX-5 RNA-binding domain is comprised of two zinc fingers that bind with high affinity to poly-U stretches. **Pagano et al. (2007)** have shown that mutation of a single amino acid in each finger (R247E and K318E) reduces MEX-5 binding affinity for poly-U by 35-fold. Using *mex-5* transgenes, **Griffin et al. (2011)** showed the finger mutations R247E and K318E also disrupt formation of the MEX-5 gradient in vivo. To determine the effect of these mutations on MEG granule assembly, we introduced R247E and K318E (hereafter referred to as ZF-) at the *mex-5* locus by CRISPR/Cas9 genome editing into an OLLAS::MEX-5 tagged line and the MEG-3::GFP line. Like MEX-5(S404A), MEX-5(ZF-) did not form a gradient and was uniformly distributed in zygotes (**Figure 2A**). In contrast to *mex-5(S404A)* zygotes, however, *mex-5(ZF-)* zygotes assembled posterior MEG-3 granules as in wild-type (data not shown). *mex-5* is partially redundant with its paralog, *mex-6* (**Schubert et al., 2000**), which is sufficient to polarize MEG granules in the absence of MEX-5 (**Figure 2—figure supplement 1**). Consistent with this redundancy, we found that depletion of *mex-6 by RNAi* in *mex-5 (ZF-)* zygotes caused MEG-3 granules to assemble throughout the cytoplasm, as in *mex-5/6(RNAi)* zygotes (**Figure 2B**). These observations suggest that *mex-5(ZF-)* is a loss-of-function allele. The loss of *mex-5* activity was not due to reduced expression as MEX-5(ZF-) was expressed at the same level as wild-type MEX-5 (**Figure 2—figure supplement 1**).

To determine whether high-affinity RNA binding is also required for MEX-5(S404A) ability to suppress MEG-3::GFP granule assembly throughout the cytoplasm, we introduced the S404A mutation by genome editing into *mex-5(ZF-)* hermaphrodites. We found that *mex-5(ZF-,S404A)* zygotes assembled posterior MEG-3::GFP granules, as is observed in *mex-5(ZF-)* and wild-type zygotes. Depletion of *mex-6* by RNAi in this background yielded zygotes with uniform MEG-3::GFP granules, as expected for a *mex-5/6* loss-of-function phenotype (**Figure 2B**). We conclude that suppression of MEG-3 granule assembly by MEX-5 depends on MEX-5's ability to bind RNA with high affinity.

## MEG-3 binds RNA in vitro

Unlike MEX-5, MEG-3 does not have a recognizable RNA-binding domain (**Wang et al., 2014**). MEG-3 contains a long predicted intrinsically-disordered region (IDR) at its N-terminus (aa1-550) followed by a region with lower predicted disorder (aa550-862) [IUPRED predictions, (**Dosztányi et al., 2005**). To determine whether MEG-3 binds RNA, we expressed and purified as His-tagged fusions full length MEG-3, MEG-3(aa1-544) (hereafter referred to MEG-3$_{IDR}$) and MEG-3(aa545-862) (hereafter referred to as MEG-3$_{Cterm}$) (**Figure 3—figure supplement 1**). We tested each for binding to poly-

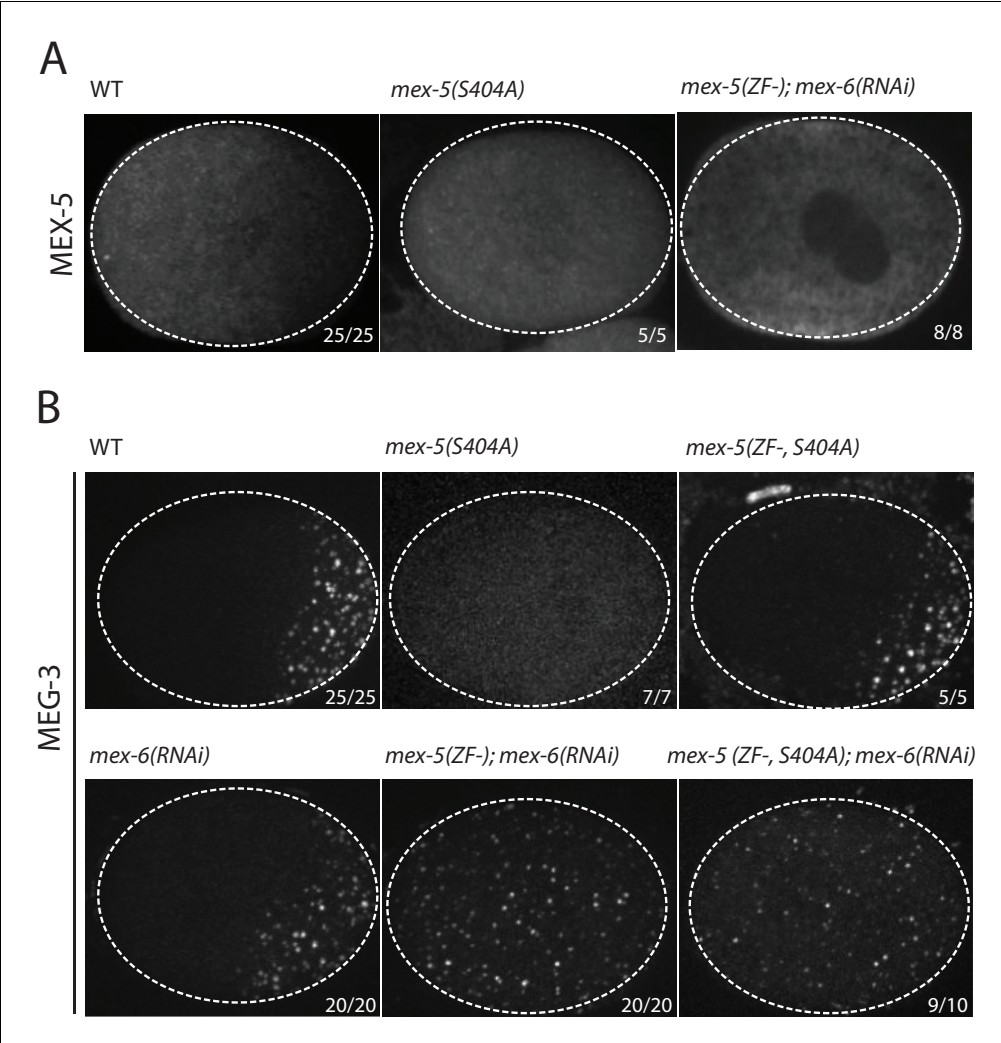

**Figure 2.** MEX-5 is necessary and sufficient to disassemble MEG-3 granules in vivo. (**A**) Photomicrographs of mCherry tagged live zygotes [wild-type and *mex-5(S404A)*] or fixed zygotes expressing MEX-5 tagged with an OLLAS epitope [*mex-5(ZF-);mex-6(RNAi)*] at pronuclear meeting to show MEX-5 localization. Wild-type MEX-5 is in an anterior-rich gradient, whereas MEX-5(S404A) and MEX-5(ZF-) are uniformly distributed. Numbers indicate number of zygotes exhibiting phenotype shown / total number of zygotes examined. (**B**) Photomicrographs of live embryos expressing MEG-3::meGFP. Genotypes at the *mex-5* locus are as indicated. Numbers indicate the numbers of zygotes examined as in A. In 1/10 *mex-5(ZF-, S404A); mex-6(RNAi)* zygotes, MEG-3 granules were asymmetric possibly due to incomplete depletion of MEX-6 by RNAi.

The following figure supplement is available for figure 2:

**Figure supplement 1.** Expression of MEX-5, and MEX-5(ZF-), MEG-3, and MEG-3$_{IDR}$.

U30 RNA using electrophoretic mobility shift assays (EMSA) and fluorescent polarization (FP) assays (*Pagano et al., 2007*). EMSA experiments revealed that MEG-3 and MEG-3$_{IDR}$ interact with poly-U30 RNA to form complexes that migrate as a discrete band during electrophoresis (*Figure 3A*). Using FP, we calculated the apparent dissociation constant ($K_{d,app}$) of MEG-3 for poly-U30 to be ~32 nM, similar to that of MEX-5 ($K_{d,app}$= ~ 29 nM) (*Pagano et al., 2007*). MEG-3$_{IDR}$ also bound RNA but with ~15 fold lower affinity ($K_{d,app}$= ~ 460 nM). MEG-3$_{Cterm}$ did not bind RNA significantly by EMSA or in the FP assay ($K_{d,app}$ > 3000 nM) (*Figure 3A*, *Figure 3—figure supplement 1*). We conclude that MEG-3 binds RNA with high affinity and that this activity resides primarily within the MEG-3 IDR, although high affinity binding also requires the MEG-3 C-terminus.

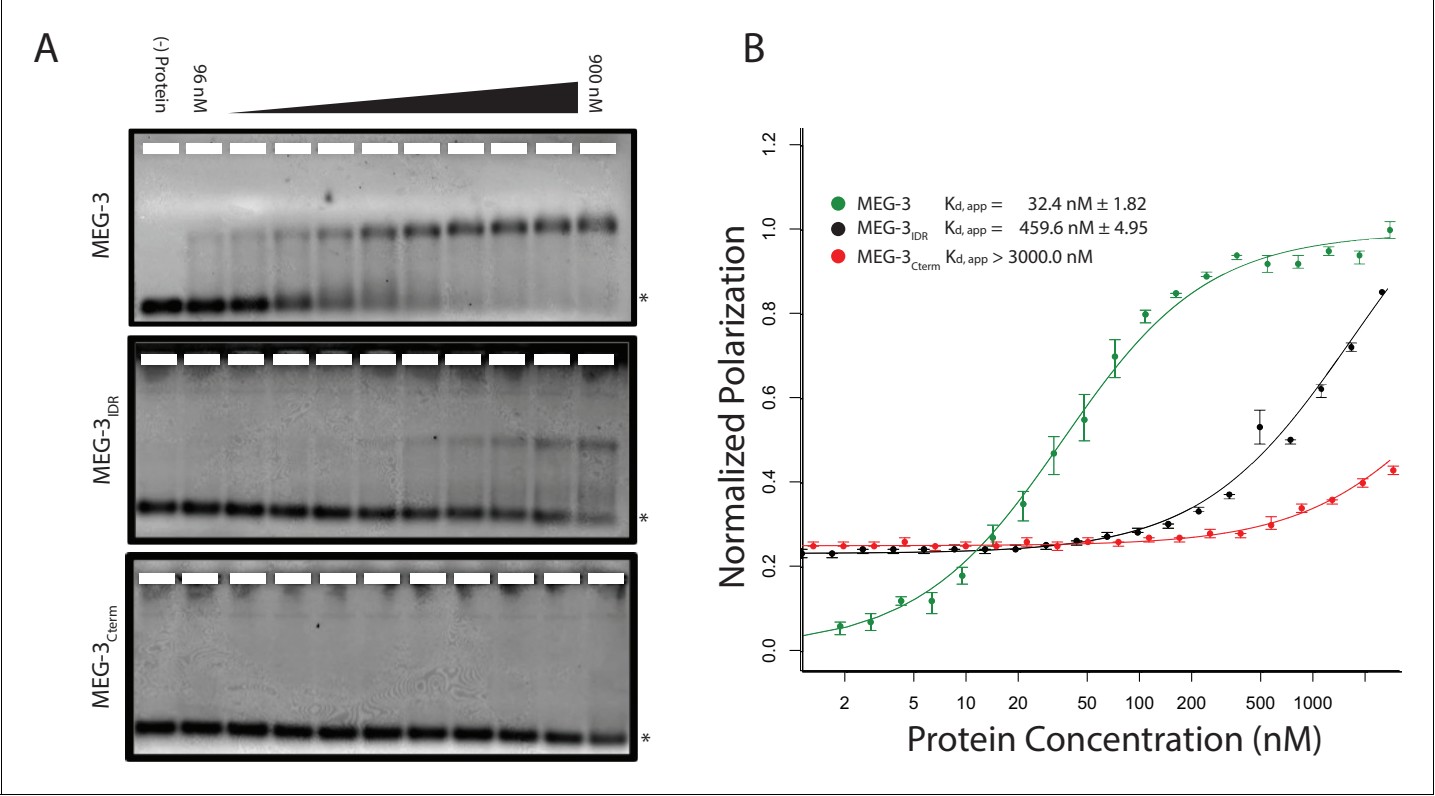

**Figure 3.** MEG-3 Binds RNA in vitro. (**A**) Binding of MEG-3 to poly-uridine 30 (poly-U30) is shown by electrophoretic mobility shift assay (EMSA) using fluorescein-labeled poly-U. EMSAs are shown for (top to bottom) full length MEG-3, MEG-3$_{IDR}$, and MEG-3$_{Cterm}$. Unbound poly-U30 is denoted by an asterisk (*). For each image shown, n $\geq$ 3 technical replicates. (**B**) Fluorescence Polarization of poly-U30 by MEG-3. Fluorescence polarization values normalized relative to saturation are shown for full length MEG-3 (green), MEG-3$_{IDR}$ (black), and MEG-3$_{Cterm}$ (red). Values represent averages of $\geq$3 technical replicates. A fit of the polarization as a function of protein concentration is plotted and used to calculate the given K$_{d,app}$. Error bars report S. E.M. Expanded graphs are shown in *Figure 3—figure supplement 1*.

The following figure supplement is available for figure 3:

**Figure supplement 1.** MEG-3 RNA binding.

To begin to explore the specificity of MEG-3 RNA binding, we challenged MEG-3/poly-U30 complexes with increasing concentrations of competitor RNAs and examined their behavior by EMSA. We found that poly-U, and to a lesser extent poly-A, were effective competitors, but not poly-C or poly-G (*Figure 3—figure supplement 1*). These observations suggest that MEG-3's affinity for RNA is affected by nucleotide composition.

## MEG-3 and MEG-3$_{IDR}$ phase separate in vitro

Concentrated (<1 μM) solutions of RNA-binding proteins containing IDRs spontaneously phase separate when switched from high to low salt (150 mM NaCl) (*Elbaum-Garfinkle et al., 2015*; *Lin et al., 2015*; *Nott et al., 2015*). We were not able to maintain high concentrations of MEG-3 or MEG-3$_{IDR}$ in solution in the presence of high salt (Materials and methods). Therefore to examine the phase separation properties of MEG-3, we used concentrated (100–320 μM) preparations maintained in 6M urea, diluted these into aqueous buffer (150 mM NaCl) and used light microscopy to immediately observe the mixture (Method). We found that MEG-3 and MEG-3$_{IDR}$ readily formed phase-separated condensates within 10 min at room temperature. Control proteins (BSA and MBP) subjected to the same treatment did not phase separate (data not shown). MEG-3 condensates were observed across a range of protein concentrations (0.5 μM to 5 μM) and became larger and more abundant with increasing protein concentration (*Figure 4—figure supplement 1*). MEG-3 and MEG-3$_{IDR}$ behaved

similarly to each other, except that in the low concentration range (<5 µM), MEG-3 formed more condensates than MEG-3$_{IDR}$, and in the high concentration range (>5 µM) MEG-3$_{IDR}$ tended to form larger condensates (*Figure 4—figure supplement 1*).

RNA can stimulate the phase transition of IDR proteins that bind RNA (*Guo and Shorter, 2015*). To determine the effect of RNA on MEG-3 phase separation, we added poly-U30 RNA to the phase separation buffer before diluting in MEG-3. We found that 0.1 µM poly-U30 was sufficient to increase the number of visible condensates especially at low MEG-3 protein concentrations (<1 µM) (*Figure 4A*, *Figure 4—figure supplement 1*). Higher concentrations of RNA increased the number of MEG-3 condensates even further. For a given concentration of poly-U30, MEG-3 formed more condensates than MEG-3$_{IDR}$ (*Figure 4A*, *Figure 4—figure supplement 1*). Addition of sub-stochiometric amounts of fluorescently tagged poly-U30 confirmed that the RNA phase separates with MEG-3 (*Figure 4—figure supplement 2*). We conclude that MEG-3 and MEG-3$_{IDR}$ have an intrinsic propensity to phase separate that can be stimulated by RNA.

## MEX-5 inhibits RNA-induced phase separation

To examine whether MEX-5 can affect MEG-3 phase separation, we purified the MEX-5 RNA-binding domain and C-terminus (aa236-468) as a His fusion (we were not able to obtain soluble full length MEX-5, Materials and methods). We pre-incubated recombinant MEX-5 with poly-U in buffer for 30 min before adding MEG-3. We found that MEX-5 strongly inhibited MEG-3 phase separation induced by RNA. Addition of excess RNA (3-fold increase) restored robust phase separation in the presence of MEX-5 (*Figure 4B*, *Figure 4C*). Additionally, MEX-5 had no effect on the phase separation of MEG-3 in the absence of RNA (*Figure 4—figure supplement 2*). These observations suggest that MEX-5 does not interfere with MEG-3 phase separation directly, but interferes with the ability of poly-U30 to induce phase separation.

## MEG-3$_{IDR}$ forms a MEX-5-dependent gradient in vivo and can be stimulated to form granules by excess RNA

Our in vitro experiments indicate that MEG-3$_{IDR}$ is sufficient to promote RNA binding and phase separation, but does so less efficiently than full length MEG-3 at low protein concentrations. To examine the behavior of MEG-3$_{IDR}$ *in vivo*, we deleted the C-terminus of MEG-3 by CRISPR/Cas9 genome editing to generate a *meg-3* allele that only expresses MEG-3$_{IDR}$ (Materials and methods, *Supplementary file 1*). We found that, like full-length MEG-3, MEG-3$_{IDR}$ is a cytoplasmic protein that redistributes into a posterior-rich gradient during polarization of the zygote (*Figure 5A*). Unlike MEG-3, however, MEG-3$_{IDR}$ did not coalesce into prominent, micron-sized granules in zygotes (*Figure 5A*). Distinct MEG-3$_{IDR}$ granules were observed starting in the 2-cell stage as MEG-3$_{IDR}$ segregates into the progressively smaller P blastomeres (*Figure 5A*). In *mex-5/6(RNAi)* zygotes, MEG-3$_{IDR}$ did not form a gradient and did not form granules (*Figure 5B*). These observations indicate that MEG-3$_{IDR}$ is partially defective in granule formation, while remaining sensitive to MEX-5/6. The loss of *meg-3* activity was not a result of reduced expression as MEG-3$_{IDR}$ was expressed at greater levels than wild-type MEG-3 (*Figure 5—figure supplement 1*).

In vitro, the weaker phase separation properties of MEG-3$_{IDR}$ at low protein concentrations can be stimulated by RNA. To determine whether excess RNA could also rescue granule formation by MEG-3$_{IDR}$ in zygotes, we blocked maternal mRNA turnover by depleting LET-711 by RNAi. LET-711 is the scaffolding component of the CCF/NOT1 deadenylase, the main deadenylase that promotes mRNA turnover in oocytes and early embryos (*DeBella et al., 2006*; *Nousch et al., 2013*). Depletion of LET-711 has been shown to increases poly-adenylation and to block the turnover of maternal *nos-2* RNA in early embryos (*Gallo et al., 2008*; *Nousch et al., 2013*). We found that MEG-3$_{IDR}$ formed numerous micron-sized granules in *let-711(RNAi)* zygotes. The MEG-3$_{IDR}$ granules and cytoplasmic gradient extended further towards the anterior compared to wild-type (*Figure 5B*). These observations suggest that, as we observed in vitro, excess RNA can overcome the inhibitory effects of MEX-5 and boost MEG-3 coalescence in vivo.

## Discussion

P granule asymmetry in *C. elegans* zygotes is a text-book example of cytoplasmic partitioning (*Strome and Wood, 1983*). In this study, we present evidence that P granule asymmetry is a direct

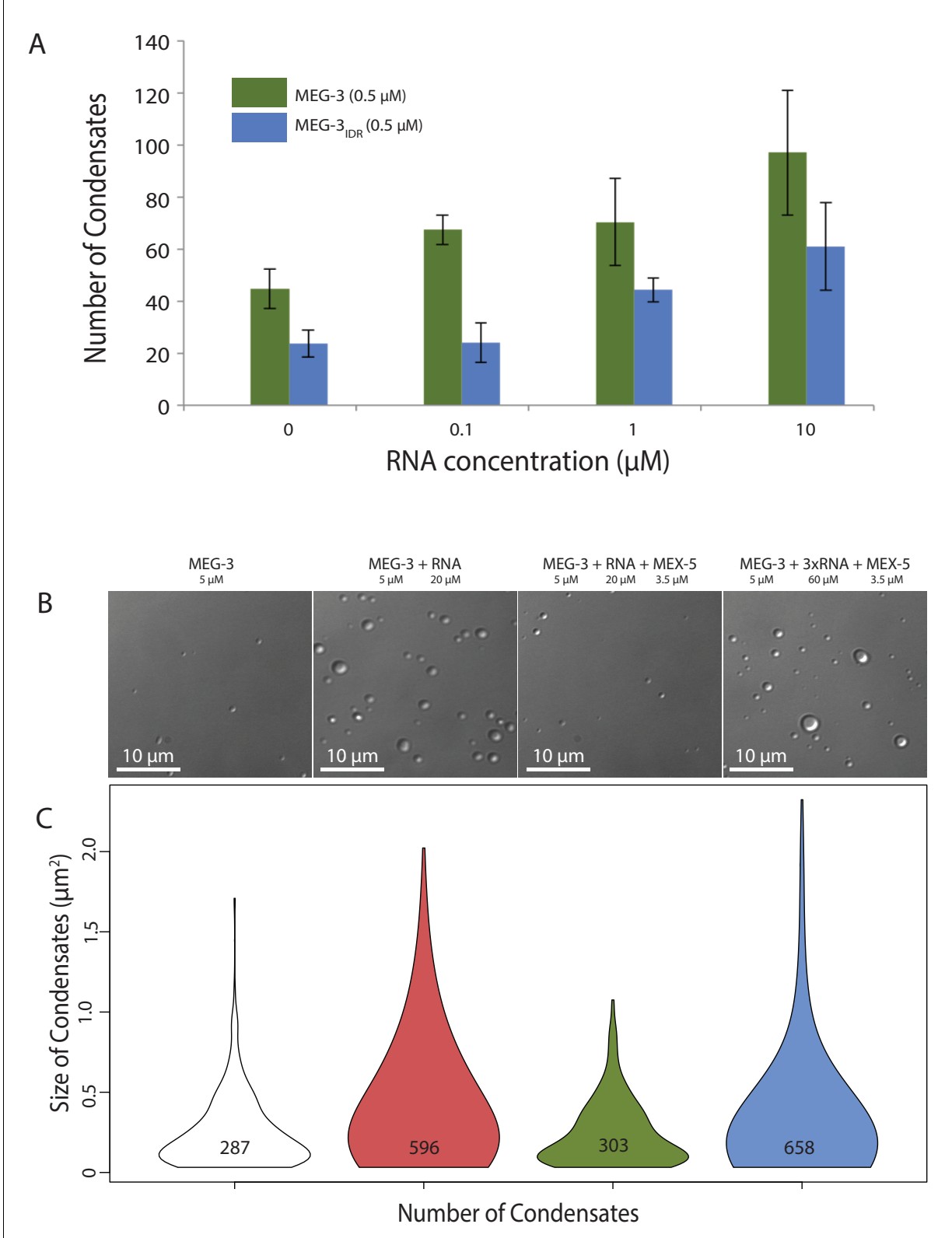

**Figure 4.** Stimulation of MEG-3 phase separation by RNA. (**A**) Bar graph showing the number of condensates formed by 0.5 µM MEG-3 or MEG-3$_{IDR}$ in the presence of increasing poly-U30. Values represent averages from three technical replicates. Error bars indicate S.E.M. (**B**) Photomicrographs of phase separation assay showing condensate formation of 5 µM full length MEG-3 incubated with poly-U RNA and/or MEX-5 as indicated.(**C**) Violin plots showing condensate size and number for each experiment represented in B. The height of the plot shows the area of condensates in µm$^2$. The width of

*Figure 4 continued on next page*

*Figure 4 continued*

the plot correlates to the proportion of condensates of that size. Numbers inside each violin plot are the total number of condensates pooled from three technical replicates for each condition. An additional control is shown in *Figure 4—figure supplement 2*.

The following figure supplements are available for figure 4:

**Figure supplement 1.** Phase Separation assay titration.

**Figure supplement 2.** Phase Separation assay controls.

consequence of an asymmetry in the distribution of the P granule scaffold MEG-3. MEG-3 localizes in a posterior-rich gradient under the control of the RNA-binding protein MEX-5, which localizes in a mirror-image, anterior-rich gradient. Our findings suggest that the MEG-3 gradient arises from an anterior-posterior gradient in RNA availability created by MEX-5. MEX-5 interferes with MEG-3 access to RNA in the anterior, which promotes MEG-3 phase separation in the posterior where MEX-5 concentration is low.

## MEG-3 is an RNA-binding protein that is stimulated by RNA to phase separate

MEG-3 contains a long N-terminal IDR but no recognizable RNA-binding domain. We have found that MEG-3 binds RNA (poly-U30) with nanomolar affinity in vitro ($K_{d,app}$ = 32 nm). The MEG-3$_{IDR}$ is essential for binding, but on its own binds with lower affinity ($K_{d,app}$ = 460 nm). One possibility is that the MEG-3 IDR extends beyond the region predicted by IUPRED (*Dosztányi et al., 2005*). The region immediately C-terminal scores close to the IUPRED cut-off (*Wang et al., 2014*), and may contribute to RNA binding. IDRs are over-represented among RNA-binding domains (*Varadi et al., 2015*; *Castello et al., 2016*). Electrostatic interactions between positively-charged amino acids and the negatively-charged RNA backbone are often invoked as a possible mechanism for RNA binding by IDRs (*Guo and Shorter, 2015*; *Basu and Bahadur, 2016*). MEG-3 is rich in basic residues, but shows a strong preference for poly-U over poly-C and poly-G, suggesting that non-charged interactions are also involved.

Several recent studies have demonstrated that RNA-binding proteins containing IDRs phase separate in aqueous solutions (*Guo and Shorter, 2015*). MEG-3 follows this paradigm: MEG-3 readily formed condensates within minutes of dilution from urea to an aqueous buffer (150 mM NaCl). MEG-3 phase separation could be stimulated by RNA: addition of poly-U30 to the phase separation buffer increased the number of MEG-3 condensates especially at 1 µM and lower protein concentrations. MEG-3$_{IDR}$ behaved similarly to full-length MEG-3, except that MEG-3$_{IDR}$ required higher concentrations of RNA to phase separate at low protein concentrations. Consistent with this in vitro behavior, MEG-3$_{IDR}$ did not form large granules in wild-type zygotes, but could be induced to do so in zygotes defective in mRNA deadenylation and turnover (RNAi depletion of the LET-711/NOT1). These observations suggest that the IDR confers on MEG-3 an intrinsic tendency for phase separation that is tunable by RNA. RNA-induced phase separation has also been observed for Whi3, a fungal RNA-binding protein, and for hnRNPA1, a stress granule protein (*Lin et al., 2015*; *Zhang et al., 2015*). In these proteins, the IDR and RNA-binding domain are distinct and RNA-induced phase separation requires both domains. It will be interesting to determine whether the MEG-3$_{IDR}$ in fact contains separable domains for phase separation and RNA binding.

## MEX-5 patterns MEG-3 by limiting access to RNA

MEX-5 has been hypothesized to regulate P granule asymmetry by creating a supersaturation gradient of critical granule component(s) along the anterior-posterior axis of the zygote (*Lee et al., 2013*). Our observations suggest that the critical component regulated by MEX-5 is RNA. MEX-5 binds RNA with nanomolar affinity ($K_{d,app}$= ~ 29 nM, *Pagano et al., 2007*) and is 10-fold more abundant than MEG-3 in embryos (*Figure 2—figure supplement 1*). In our in vitro phase separation assay, the MEX-5 RNA-binding domain was sufficient to inhibit RNA-induced phase separation of MEG-3. Similarly, in vivo, uniform MEX-5 was sufficient to inhibit MEG-3 granule assembly

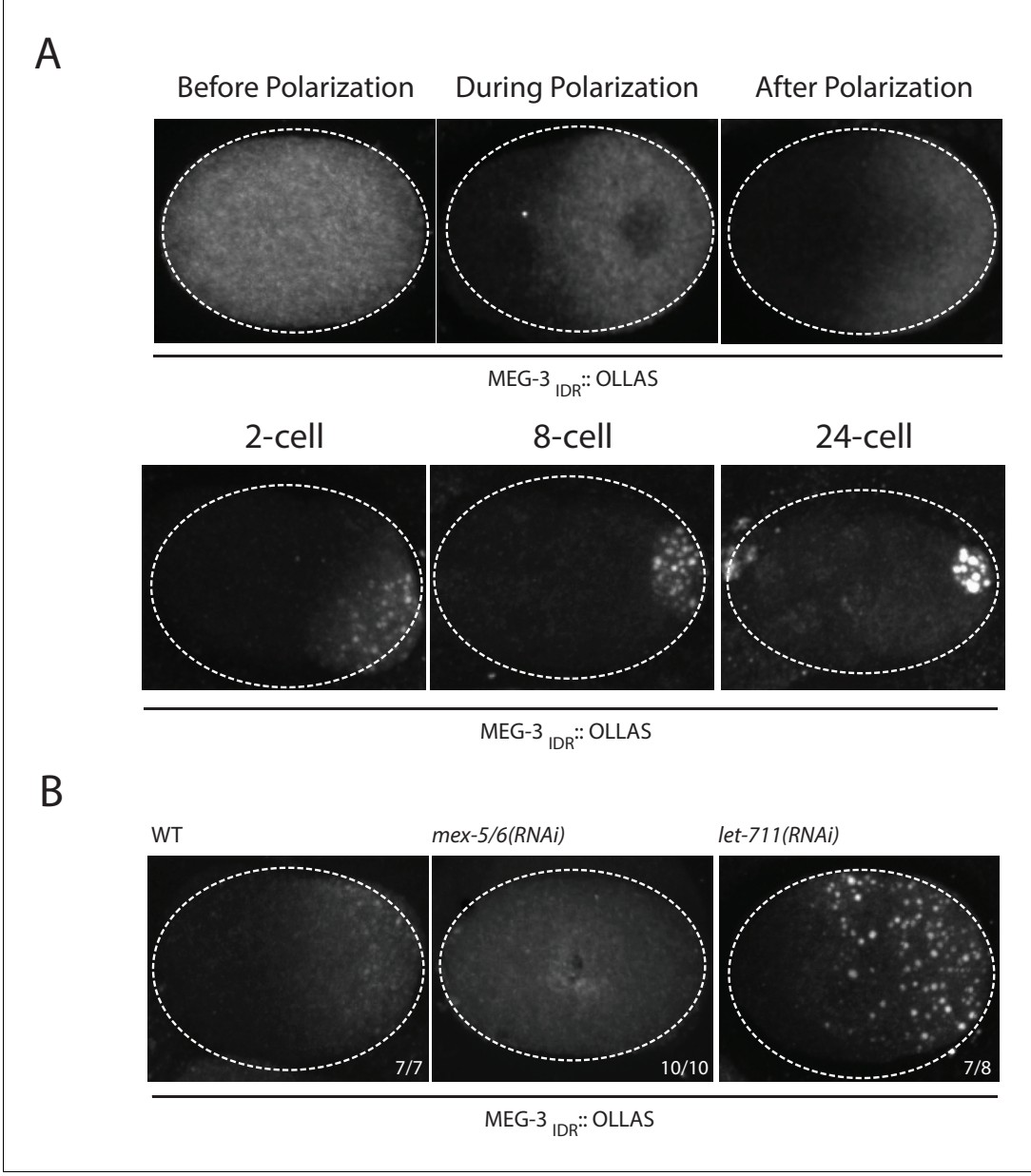

**Figure 5.** Coalescence of MEG-3$_{IDR}$ can be stimulated by blocking mRNA turnover in vivo. (**A**) Photomicrographs of fixed embryos expressing MEG-3$_{IDR}$ tagged with OLLAS epitope. First row are zygotes (one-cell stage) and second row are later stage embryos as indicated. (**B**) Photomicrographs of fixed zygotes expressing MEG-3$_{IDR}$ tagged with OLLAS epitope. Genotypes are indicated above each embryo (left to right: wild-type, *mex-5/6(RNAi)*, and *let-711(RNAi)*). Numbers indicate number of zygotes exhibiting phenotype shown / total number of zygotes examined. In 1/8 *let-711(RNAi)* zygotes, MEG-3$_{IDR}$ formed granules but these were smaller and confined to the posterior half of the zygote, possibly due to incomplete depletion of *let-711*.

The following figure supplement is available for figure 5:

**Figure supplement 1.** MEG-3$_{IDR}$ in vivo.

throughout the cytoplasm and this activity was disrupted by mutations that lower MEX-5's affinity for RNA. Together, these results support a model where MEX-5 suppresses MEG-3 phase separation by limiting MEG-3's access to RNA.

How MEX-5 regulates MEG-3's access to RNA in vivo remains to be determined. As suggested by the in vitro observations, MEX-5 could compete directly with MEG-3 for RNA binding. Alternatively, MEX-5 could function indirectly by recruiting factors to MEG-3/RNA complexes that reduce MEG-3's affinity for RNA. The MEX-5 gradient arises as a consequence of phosphorylation by the posteriorly-enriched kinase PAR-1, which decreases the size of MEX-5 complexes and increases MEX-5's diffusion rate (*Griffin et al., 2011*). One possibility is that phosphorylation by PAR-1 prevents MEX-5 from binding RNA creating a pool of 'MEX-5-free' RNA available to phase separate with MEG-3 in the posterior cytoplasm.

While MEX-5 was sufficient to inhibit the RNA-induced phase separation of MEG-3 in vitro, it was not able to reverse it. Addition of MEX-5 to preformed MEG-3 condensates was not sufficient to dissolve them (data not shown). In contrast, in vivo, MEG-3 granules dissolve in the anterior cytoplasm as MEX-5 concentration increases (*Figure 1A*). We showed previously that, in vivo, P granule dynamics are also regulated by phosphorylation. The MBK-2 kinase and the PP2A phosphatase promote granule disassembly and assembly, respectively (*Wang et al., 2014*). One possibility is that cycles of MEG phosphorylation and de-phosphorylation promote cycles of granule disassembly and re-assembly throughout the cytoplasm. By limiting RNA access, MEX-5 inhibits the re-assembly of MEG-3 condensates in the anterior cytoplasm freeing more MEG-3 for granule assembly in the posterior cytoplasm.

MEX-5's high affinity for poly-U stretches, which are present in 91% of *C. elegans* 3' UTRs (*Pagano et al., 2007*), suggests that MEX-5 interacts with most mRNAs in zygotes and thus could function as a general 'mRNA sink'. Consistent with this view, the MEX-5 gradient patterns the distribution of three other RNA-binding proteins that, like MEG-3, form posterior-rich gradients (*Wu et al., 2015*). The observation that blocking mRNA turnover stimulates MEG-3 coalescence into macroscopic granules is consistent with the idea that the mRNA pool accessible to MEG-3 is limiting in zygotes. A limiting mRNA pool has also been suggested to regulate the balance of P bodies and stress granules in cells (*Buchan and Parker, 2009*). We propose that regulated access to mRNAs, combined with RNA-induced phase separation of key scaffolding proteins, may be a general mechanism for controlling the formation of RNA granules in space and time.

## MEG-3 and MEG-4 scaffold P granules in zygotes

A recent theoretical study has suggested that MEX-5 patterns P granules by regulating the phase separation of a different P granule protein: PGL-3 (*Saha et al., 2016*). This model is based on the assumption that PGL-3 is essential to assemble P granules and on three in vitro observations: (1) PGL-3 binds RNA using a C-terminal RGG domain, (2) RNA-binding via the RGG domain stimulates phase separation of PGL-3 and (3) MEX-5 can interfere with the formation of PGL-3/RNA condensates. Using estimates for the in vivo concentrations and diffusion rates of PGL-3, MEX-5 and RNA, Saha et al. built a mathematical model that simulates a competition for RNA between MEX-5 and PGL-3. In the simulation, introduction of the MEX-5 gradient is sufficient to dissolve PGL-3 granules in the anterior cytoplasm. This model is inconsistent with several in vivo observations. First, *Hanazawa et al. (2011)* have reported that PGL-3 lacking the RGG domain still forms asymmetric granules in zygotes. Second, the formation of the MEX-5 gradient does not correlate temporally with, and is not sufficient for, PGL-3 asymmetry in vivo (*Figure 1—figure supplement 1*). Third, PGL-3 and its paralog PGL-1 are not required to assemble or localize MEG granules in vivo (*Figure 1C*, *Figure 1—figure supplement 1*). The in vivo data demonstrate that, contrary to the model proposed by *Saha et al. (2016)*, PGL-3 does not require RNA binding to localize to the granules, does not sense the MEX-5 gradient directly, and does not drive granule assembly or asymmetry. The in vivo data indicate instead that MEG-3/4 are responsible for P granule asymmetry and recruitment of PGL-3 to P granules. In vitro, MEG-3 binds to PGL-1 and PGL-1 binds to PGL-3 (*Hanazawa et al., 2011*; *Wang et al., 2014*), raising the possibility that the MEGs recruit the PGLs by direct protein-protein interactions. We showed previously that MEG-3 overlaps but does not co-localize perfectly with PGL-3 in P granules (*Wang et al., 2014*). Formation of stable PGL assemblies in embryos also requires LAF-1, a DEAD-box RNA helicase that phase separates in vitro independently of RNA (*Elbaum-Garfinkle et al., 2015*). P granules therefore appear to comprise multiple phases, each with distinct properties and components that have an affinity for one another but do not fully mix. We suggest that MEG-3 and MEG-4 form the main RNA-dependent phase of P granules. The MEG phase functions as a scaffold that recruits other P granule components to build multi-

phasic assemblies. An important question will be to determine the physical and biochemical properties of the MEG phase. The conclusion that P granules are liquid droplets was based mostly on observations of the PGL phase (*Brangwynne et al., 2009*). Whether the MEG phase is also liquid or is a more ordered gel-like or solid condensate remains to be determined. How the MEG scaffold specifies the unique RNA/protein composition of P granules will be an exciting area for future inquiry.

## Materials and methods

### CRISPR genome editing

*C. elegans* was cultured according to standard methods at 20°C (*Brenner, 1974*). Genome editing was performed using CRISPR/Cas9 as described in *Paix et al. (2015)*. Alleles used in this study are listed in *Supplementary file 1*.

### RNA mediated interference (RNAi)

RNAi knock-down experiments were performed by feeding on HT115 bacteria (*Timmons and Fire, 1998*). Feeding constructs were obtained from the Ahringer or Openbiosystems libraries and transformed into HT115 bacteria. pL4440 was used as a negative control empty feeding vector. Bacteria were grown at 37°C in LB + ampicillin (100 µg/mL) for 5 hr, induced with 5 mM IPTG for 45 min, plated on NNGM (nematode nutritional growth media) + ampicillin (100 µg/mL) + IPTG (1 mM), and grown overnight at room temperature. Embryos isolated by bleaching from gravid hermaphrodites were added to the RNAi plates and transferred to fresh plates as L4 larvae before examination of their progeny. All RNAi experiments were performed at 20°C.

### Protein expression and purification

All purifications were performed using an AKTA pure FPLC protein purification system (GE Healthcare Silver Spring, MD).

#### Purification of MEG-3 and MEG-3$_{IDR}$

MEG-3 (aa1-862), MEG-3$_{IDR}$ (aa1-544) fused to an N-terminal 6XHis tag in pET28a were expressed in Rosetta (DE3) cells at 16°C in LB + ampicillin (100 µg/mL) to an OD600 of ~0.4 and induced with 0.4 mM isopropyl $\beta$-D-1-thiogalactopyranoside at 16° C for 16 hr. Cells were resuspended in Buffer A (20 mM HEPES, 500 mM NaCl, 20 mM Imidazole, 10% (vol/vol) glycerol, 1% Triton-X100, 6M Urea, 6 mM $\beta$ME, pH7.4) with added protease inhibitors and TCEP, lysed by sonication, spun at 13,000 rpm for 25 min, and incubated overnight at 4C. Lysate was passed over a His Prep FF 16/10 column (GE Healthcare Silver Spring, MD). Bound protein was washed with Buffer B (20 mM HEPES, pH 7.4, 1M NaCl, 25 mM Imidazole, 10% (vol/vol) glycerol, 6M urea, 6 mM $\beta$ME) and eluted in Buffer C (20 mM HEPES, pH 7.4, 1M NaCl, 250 mM Imidazole, 10% (vol/vol) Glycerol, 6M Urea, 6 mM $\beta$ME). After each purification, aliquots of the peak elution fraction were run on 4–12% Bis Tris gels, and stained with Simply Blue Safe Stain (ThermoFisher Waltham, MA). Proteins were concentrated to a final concentration of 100–320 µM in elution Buffer C. For use in RNA binding assays, proteins were dialyzed into storage buffer B (25 mM Hepes, pH 7.4, 1M NaCl, 6 mM $\beta$ME, 10% (vol/vol) glycerol) and stored at −80° C.

#### Purification of MEG-3$_{Cterm}$

MEG-3 (545–862) was purified as above and also natively using the same protocol without urea. MEG-3$_{Cterm}$ purified under native conditions was soluble in aqueous buffer even at high concentrations (>1 uM) and was used for RNA-binding assays.

#### Dialysis of MEG-3

MEG-3 that was utilized in RNA binding experiments was step dialyzed out of urea, into 4.5M Urea, 3M Urea, 1.5M Urea, and 0M Urea in MEG-3 Storage Buffer (25 mM HEPES, pH 7.4, 1M NaCl, 6 mM $\beta$ME, 10% Glycerol). The protein was aliquoted and flash frozen in MEG-3 Storage Buffer and stored at −80C.

## Purification of MEX-5

MEX-5(aa236-468) was purified as an N-terminal 6xHis:MBP fusion expressed in Rosetta (DE3) competent cells. Cells were grown at 37° C in LB + ampicillin (100 µg/mL) to an OD600 of ~0.4, before induction with 0.2 mM isopropyl $\beta$-D-1-thiogalactopyranoside and 100 µM zinc acetate at 16° C for 16 hr. Cells were resuspended in lysis buffer (20 mM Tris-HCl, pH 8.3, 200 mM NaCl, 20 mM imidazole, 10% (vol/vol) glycerol, 1 mM TCEP, 100 µM zinc acetate, Roche complete EDTA-free protease inhibitor tablet), lysed by sonication and pelleted at 10,000 rcf for 15 min. The supernatant was passed over a HisTrap HP column (GE Healthcare Silver Spring, MD) and washed with wash buffer (20 mM Tris-HCl, pH8.3, 800 mM NaCl, 20 mM imidazole, 10% (vol/vol) glycerol, 100 µM zinc acetate,1 mM TCEP). Column was eluted using elution buffer (20 mM HEPES, pH 8.3, 500 mM NaCl, 250 mM imidazole, 10% (vol/vol) glycerol, 100 µM zinc acetate). Elution fractions were pooled and run over a HiTrap Heparin HP column (GE Healthcare Silver Spring, MD). Column was then washed in wash buffer B (20 mM Hepes, pH 8.4, 200 mM NaCl) and eluted using a gradient of wash buffer B and elution buffer B (20 mM Hepes, pH 8.4, 1.5 M NaCl, 100 µM zinc acetate). Elutions were pooled and dialyzed into storage buffer as in *Pagano et al. (2007)* (20 mM Tris, pH 8.3, 20 mM NaCl, 100 µM zinc acetate, 10% (vol/vol) glycerol). Protein concentration was determined by measuring absorbance at 280 nm as in *Pagano et al. (2007)* and stored at −80°C.

## Immunostaining

Adult worms were placed into M9 salt solution on epoxy autoclavable slides (thermofisher Waltham, MA) and squashed with a coverslip to extrude embryos. Slides were frozen by laying on pre-chilled aluminum blocks for 20 min (chilled using dry ice). Embryos were permeabilized by freeze-cracking (removal of coverslips from slides) followed by incubation in methanol at −20°C for >15 min, and in acetone (pre-chilled at −20°C) at room temperature for 10 min. Slides were blocked in PBS-Tween (0.1%) BSA (0.5%) for 15 min x 2, and incubated with 50 ul primary antibody overnight at 4°C in a humid chamber. Antibody dilutions (in PBST/BSA): K76 (1:10, DSHB), Rat α OLLAS-L2 (1:200, Novus Biological Littleton, CO), mouse α FLAG (1:500, Sigma St. Louis, MO). Secondary antibodies were applied for 2 hr at room temperature.

## Confocal microscopy

Fluorescence microscopy was performed using a Zeiss Axio Imager with a Yokogawa spinning-disc confocal scanner. Images were taken and stored using Slidebook v 6.0 software (Intelligent Imaging Innovations) using a 63x objective. For live imaging, embryos were dissected from adult hermaphrodites in M9 salt solution and mounted onto 3% agarose pads. All embryo images are z stack maximum projections using a z step size of 1 µm, spanning the entire width of the embryo.

## Quantification of MEG-3::meGFP fluorescence from confocal images

Equally normalized time-lapse images were quantified using Slidebook v 6.0. Average fluorescence intensity relative to area of anterior (60%) and posterior (40%) of zygote were quantified and average fluorescence intensity relative to area of background (outside of zygote) was subtracted from each of these values. For each time-point, anterior and posterior fluorescence were expressed as fractions of total fluorescence and then normalized to $T_0$ (14 min prior to mitosis). Final values represent average of three embryos. Error bars show standard deviation of the mean.

## Electrophoretic mobility shift assay (EMSA)

EMSA were carried out as described in *Pagano et al. (2007)*. Reactions consisted of 50 nM 3' Fluorescein-labeled RNA oligonucleotides (Dharmacon -GE Lifesciences Silver Springs, MD) incubated with protein for 2 hr or more at room temperature or for 30 min followed by 2 hr incubation with unlabeled competitor. Samples were run on 1% agarose gel in 1x TAE. Gels were scanned immediately using typhoon FLA-9500 with blue laser at 473 nm.

## Fluorescence polarization assay

Equilibration reactions were performed using same protocol as for EMSAs. Reactions were transferred to 384 well microplates (Greiner Bio-One Monroe, NC). The apparent fluorescence polarization was determined using a Clariostar monochromator microplate reader with fluorescein-sensitive

filters and polarizers. Polarization values were normalized relative to saturation polarization value. For each experiment, values of three reads were averaged. Average values and standard errors from at least three technical replicates were calculated and plotted against each protein concentration. These data were fit to a quadratic equation (*Equation 1*, where b is the base polarization, m is the maximum polarization, R is the labeled nucleic acid concentration, and P is the total protein concentration) as in (*Pagano et al., 2011*), to calculate the apparent dissociation constant. The reported values are the dissociation constants calculated using the polarization values averaged from all technical replicates. The reported errors are standard error values calculated from the dissociation constant of each individual technical replicate.

$$\phi = b + (m - b) \times \left[ \frac{R + P + K_{d,app} - \sqrt{\left(R + P + K_{d,app}\right)^2 - (4RP)}}{2R} \right] \quad (1)$$

## Phase separation assay

His::tagged MEG-3 fusions were quickly diluted out of urea into condensation Buffer (25 mM HEPES, pH7.4, NaCl adjusted to a final concentration of 150 mM) in the presence and absence of poly-U30 RNA. Dilutions were performed by adding buffer to protein in low-binding siliconized Eppendorf tubes and mixing briefly by pipetting. The reaction was either spun at 3000 rpm for 2 min or transferred directly into 35 mm glass bottom dish (Cat. No. P35G-1.5–14 C MatTek Corp Ashland, MA) for imaging. Phase separation assays with MEX-5 were performed by pre-incubating 3.5 µM 6xHis:: MBP::MEX-5 protein (dialyzed into condensation buffer) with condensation buffer and poly-U30 RNA for 30 min before diluting in MEG-3. Differential interference contrast (DIC) images were obtained on an Olympus inverted microscope, using a 100X objective. Images were taken and stored using Slidebook v 4.0 and 5.0. All images are a single focal plane focused on the slide surface. For each phase separation experiment, we took three separate images of an $80 \times 80$ micron field and counted the condensates using Image J64. To recognize condensates, background was subtracted from each image using a rolling ball radius of 10 pixels, a pixel brightness threshold was set to 15– 255. Remaining pixels were smoothed three times and size and number of objects greater than $0.032 \ \mu m^2$ were quantified. Quantifications were manually verified for each image used. At least three technical replicates were quantified for each condition. Average number of condensates and S.E.M were calculated using all technical replicates.

## Western blots

Western blots were performed by running worm lysates on 7% Tris Acetate SDS PAGE precast gels (Bio-Rad Hercules, CA). Protein was transferred to nitrocellulose membrane which was pre-blocked in 5% Milk diluted in PBS-Tween (0.1%) for 5 min (three times). The membrane was then incubated with Primary antibody for at least 18 hr at 4° C or 2 hr at room temperature. Membranes were washed and blocked in 5% milk for 5 min (three times) and incubated with secondary HRP conjugated antibody for 45 min at room temperature. Membranes were washed in 5% milk for 5 min (two times) and PBST for 5 min (one time). The membranes were then exposed to ECL substrate for 1 min and then exposed to film. Primary antibody dilutions (in 5% Milk PBST): Rat $\alpha$ OLLAS-L2 (1:1000, Novus Biological Littleton, CO), Mouse $\alpha$ Tubulin (1:1000, Sigma St. Louis, MO).

## Technical v biological replicates

All in vivo biological replicates refer to experiments performed on independently isolated hermaphrodites (in the case of mutants, this refers to separate strains isolated from independent editing events) or independently treated hermaphrodites (in the case of RNAi, this refers to wild-type worms exposed to independent RNAi treatments). In vitro biological replicates refer to experiments performed with independently purified protein preps. All in vivo technical replicates refer to observations in the same strain from separate zygotes. In vitro technical replicates refer to separate experiments performed using the same purified protein preps.

## Acknowledgements

We thank Erik Griffin for discussions, the CGC (USA) for strains, the Berger lab (JHU) for reagents and use of their microplate reader, Alex Paix for the *glh(ax3064)* allele, and the JHMI microscope facility for instruction. Research in the Seydoux lab is supported by R01 HD37047 from the National Institute of Health. G Seydoux is an Investigator of the Howard Hughes Medical Institute.

## Additional information

### Funding

| Funder | Grant reference number | Author |
|---|---|---|
| NIH Office of the Director | R01 HD37047 | Deepika Calidas<br>Helen Schmidt<br>Tu Lu |
| Howard Hughes Medical Institute | | Geraldine Seydoux |

The funders had no role in study design, data collection and interpretation, or the decision to submit the work for publication.

### Author contributions

JS, DC, HS, Conception and design, Acquisition of data, Analysis and interpretation of data, Drafting or revising the article; TL, Conception and design, Acquisition of data, Analysis and interpretation of data; DR, Acquisition of data, Contributed unpublished essential data or reagents; GS, Conception and design, Analysis and interpretation of data, Drafting or revising the article

### Author ORCIDs

Jarrett Smith, http://orcid.org/0000-0001-6354-6551
Deepika Calidas, http://orcid.org/0000-0002-0859-5390
Helen Schmidt, http://orcid.org/0000-0002-3449-2790
Tu Lu, http://orcid.org/0000-0002-5697-300X
Geraldine Seydoux, http://orcid.org/0000-0001-8257-0493

## Additional files

### Supplementary files

• Supplemental file 1. Strains used in this study. All strains were generated in this study by genome editing or crossing. No transgenic lines were used. Independent edits displayed the same phenotypes. The *mex-5(S404A)* lines could not be maintained due to semi-dominant maternal-effect sterility (91.6%) and recessive maternal-effect lethality (100%).

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
