## [Decision Letter]

Thank you for submitting your article "Spatial patterning of P granules by RNA-induced phase separation of the intrinsically-disordered protein MEG-3" for consideration by *eLife*. Your article has been reviewed by three peer reviewers, and the evaluation has been overseen by Julie Ahringer as the Reviewing Editor and Vivek Malhotra as the Senior Editor. The reviewers have opted to remain anonymous.

The reviewers have discussed the reviews with one another and the Reviewing Editor has drafted this decision to help you prepare a revised submission. A compiled list of essential revisions is given below. Full reviews are appended at the end of this letter for your information.

Summary:

The article by Smith, Seydoux and colleagues provides new insight into the organizational principles of P-granule assembly in *C. elegans*. The authors show P granule assembly occurs stepwise and critically depends on the function of Mex 5/6 and Meg3/4. They further demonstrate that Mex-5 needs to be phosphorylated and have intact Zn fingers that can bind to poly U stretches to generate *meg-3/4* gradient. The most interesting aspect of the study is the analysis of the ability of Meg3 (4) to bind RNA primarily through its N-terminal IDR domain. Interestingly in vitro the affinity for polyU is similar between Mex5 and Meg3, suggesting a competition between these two proteins for RNA. The authors demonstrate in vitro phase transitions of purified IDR and full length Meg3, that this is increased by polyU RNA and can be competed by Mex5 binding to RNA. in vivo the Meg3 IDR is able to form granules however at a later stage or under increased RNA levels. Overall this is a very interesting study that outlines how non-sequence specific RNA interactions with IDR domains can facilitate phase transitions.

Essential revisions:

1) Quantification of granule number and size to back up qualitative claims. —The images in Figure 1 and Figure 1—figure supplement 1 on the influence of *pgl* mutants on Meg 3 granules appear quite different. In Figure 1 there seems to be a weakening of localization and granule size while the granules seem very similar to wildtype in Figure 1—figure supplement 1. Could the authors develop a more quantitative description for both sizes and numbers? Granule size seems also differently affected in *mex 5/6* mutants for Pgl or Meg granules, again is this just the images presented or a consistent difference.

2) This careful study is in striking conflict to a study published by Hyman and colleagues jut last month in Cell. In this study a role for Pgl proteins akin to the function of Meg3/4 proposed here is populated. The authors briefly discuss this in the present manuscript, but a more complete discussion would be helpful. In particular, in vitro Pgl proteins can phase separate and recruit other granule components but apparently in vivo granule formation depends on Meg3/4. Can this be related to a functional significance for germ line development?

3) The findings in this paper are a significant departure from the conclusions this same group published two years ago in *eLife* showing that MEG-3 is a substrate of the kinase MBK-2 and the phosphatase PPTR-1/2 (Wang, et al.). In the Wang study, phosphorylation of MEG-3 by MBK-2 in the anterior of the *C. elegans* zygote promoted P-granule disassembly, while dephosphorylation by PPTR-1/2 promoted P-granule assembly. Attempts to reconcile these findings with the new discovery that asymmetric MEG-3 distribution and assembly now depends on an mRNA-binding competition with MEX-5 need to be included.

4) A central argument that distinguishes this paper from Saha et al. is that MEX-5 drives P-granule disassembly through MEG-3 instead of PGLs. Two findings support this. The first is that posterior enrichment of MEG-3::OLLAS is observed before posterior enrichment of PGL-1. PGL-3 is not examined and should be as the Saha et al. paper showed a specific PGL-3 response. There are available PGL-3 antibodies to test this. The second finding is that MEG-3 is still enriched in the posterior of *pgl-1/3* RNAi embryos; however, the degree of knock down is not confirmed so PGL-1/3 may still be present. Co-stains of MEG-3/4 (anti-FLAG/OLLAS) with anti-PGL-1/3 in pgl-1/3(RNAi) embryos should be straight forward with existing antibodies.

5) Findings presented in Figure 2 need clarification. It is not immediately clear that anterior MEX-6 is sufficient to prevent MEG-3 granules in the text. Nomenclature and labels in Figure 2 need improved to better sync with the text.

6) It is not obvious as to why the ZF- mutated MEX-5 (R247E and K318E) also does not form a gradient, even though it has the necessary serine at position 404.

7) Figure 1 – should specify for clarity what timepoint this is (before, during, or after polarization).

*Reviewer #1:*

The article by Smith, Seydoux and colleagues provides new insight into the organizational principles of P-granule assembly in *C. elegans*. The authors show P granule assembly occurs stepwise and critically depends on the function of Mex 5/6 and Meg3/4. They further demonstrate that Mex-5 needs to be phosphorylated and have intact Zn fingers that can bind to poly U stretches to generate *mg3/4* gradient. The most interesting aspect of the study is the analysis of the ability of Meg3 (4) to bind RNA primarily through its N-terminal IDR domain. Interestingly in vitro the affinity for polyU is similar between Mex5 and Meg3, suggesting a competition between these two proteins for RNA. The authors demonstrate in vitro phase transitions of purified IDR and full length Meg3, that this is increased by polyU RNA and can be competed by Mex5 binding to RNA. in vivo the Meg3 IDR is able to form granules however at a later stage or under increased RNA levels.

Many of the results with regard to reciprocal gradient formation of *mex5/6* and Meg3/4, are not new, but in this study the use of crispr mutations and tagging of the endogenous locus makes the analysis more rigorous and less likely to be conflicted by trans gene expression. Overall this is a very interesting study that outlines how non-sequence specific RNA interactions with IDR domains can facilitate phase transitions. in vivo, granules are not composed of single molecules and the authors have shown in previous studies that P granules are structurally organized. How this organization is affected by the non-sequence specific RNA interactions is not clear.

Given the interest in phase transitions and granule formation including plenty of controversies, this is a refreshingly solid and technically particularly careful study which in my mind seems perfect for *eLife*.

1) Quantification of granule number and size. The images in Figure 1 and Figure 1—figure supplement 1 on the influence of *pgl* mutants on Meg 3 granules appear quite different. In Figure 1 there seems to be a weakening of localization and granule size while the granules seem very similar to wildtype in Figure 1—figure supplement 1. Could the authors develop a more quantitative description for both sizes and numbers? Granule size seems also differently affected in *mex 5/6* mutants for Pgl or Meg granules, again is this just the images presented or a consistent difference.

2) Biological role of Meg protein phase transitions. What is the phenotype of Meg3 IDR with regard to germ line development in a *meg4* mutant? Meg 3 and 4 are redundant is this at the level of function or phase transition. For example, can Meg3 and Meg4 form hetero-polymers in vivo?

3) This careful study is in striking conflict to a study published by Hyman and colleagues jut last month in Cell. In this study a role for Pgl proteins akin to the function of Meg3/4 proposed here is populated. The authors briefly discuss this in the present manuscript, but a more complete discussion would be helpful. In particular, in vitro Pgl proteins can phase separate and recruit other granule components but apparently in vivo granule formation depends on Meg3/4. Can this be related to a functional significance for germ line development?

*Reviewer #2:*

In this paper, Smith, et al., address the poorly understood process of cytoplasmic RNA-granule phase separation during development. Previous work has shown that a series of asymmetric protein gradients are established after fertilization, culminating in the anterior enrichment of an RNA-binding protein called MEX-5 in the *C. elegans* zygote prior to mitosis. MEX-5 is necessary to block phase separation of P granules, but posterior MEX-5 is destabilized so P granules phase separate and segregate with the nascent germline. P granules consist of a heterogeneous mix of MEG, PGL, and GLH proteins (among other things), but the link between MEX-5 and P-granule proteins was unknown. Recently, a separate group (Saha, et al. Cell 2016) reconstituted P granule-like droplets in vitro, demonstrated that competition between PGL-3 and MEX-5 for mRNA can regulate droplet formation, and used theory to show that a MEX-5 gradient can drive PGL-3 and P-granule assembly in vivo. Smith et al. complement the findings of Saha et al., confirming that MEX-5 does act as an RNA sink to prevent phase separation, but in contrast they also demonstrate that the mRNA binding competition in the zygote is between MEX-5 and MEG-3 instead of MEX-5 and PGL-3. These findings make a significant contribution to what drives phase separation and P-granule assembly in the cytoplasm.

1) The findings in this paper are a significant departure from the conclusions this same group published two years ago in *eLife* showing that MEG-3 is a substrate of the kinase MBK-2 and the phosphatase PPTR-1/2 (Wang, et al.). In the Wang study, phosphorylation of MEG-3 by MBK-2 in the anterior of the *C. elegans* zygote promoted P-granule disassembly, while dephosphorylation by PPTR-1/2 promoted P-granule assembly. Attempts to reconcile these findings with the new discovery that asymmetric MEG-3 distribution and assembly now depends on an mRNA-binding competition with MEX-5 need to be included.

2) A central argument that distinguishes this paper from Saha et al. is that MEX-5 drives P-granule disassembly through MEG-3 instead of PGLs. Two findings support this. The first is that posterior enrichment of MEG-3::OLLAS is observed before posterior enrichment of PGL-1. PGL-3 is not examined and should be as the Saha et al. paper showed a specific PGL-3 response. There are available PGL-3 antibodies to test this. The second finding is that MEG-3 is still enriched in the posterior of pgl-1/3 RNAi embryos; however, the degree of knock down is not confirmed so PGL-1/3 may still be present. Costains of MEG-3/4 (anti-FLAG/OLLAS) with anti-PGL-1/3 in pgl-1/3(RNAi) embryos should be straight forward with existing antibodies.

3) Findings presented in Figure 2 need clarification. It is not immediately clear that anterior MEX-6 is sufficient to prevent MEG-3 granules in the text. Nomenclature and labels in Figure 2 need improved to better sync with the text.

*Reviewer #3:*

In this manuscript by Smith et al. 2016, the nature of RNA dependent phase separation in relation to P granules in the early *C. elegans* embryo is studied. Recent work by these authors demonstrated that a group of intrinsically disordered, serine-rich proteins (specifically the MEG proteins) regulates the dynamics of P granule assembly and disassembly. Here, with a combination of in vivo and in vitro work, the authors describe a model in which RNA binding competition between MEX-5 and MEG-3 results in P granule asymmetry. in vivo, the authors first show the opposing localization patterns of MEX-5 and MEG-3, and through mutational studies show that MEX-5 gradients drive MEG-3 granule asymmetry. in vitro, the authors show that purified MEG-3 (and its disordered region) can phase separate, and that this phase separation is greatly enhanced by addition of RNA through MEG-3 RNA binding. Finally, it is shown that addition of MEX-5 in vitro inhibits phase separation of MEG-3. These findings dovetail nicely with recent work showing a similar phenomenon with MEX-5 and PGL-3 (Saha et al. 2016). Overall, this study provides a clear and logical presentation of experimental work to support the majority of its conclusions, and I fully support its publication in *eLife*.

[Editors' note: further revisions were requested prior to acceptance, as described below.]

Thank you for resubmitting your work entitled "Spatial patterning of P granules by RNA-induced phase separation of the intrinsically-disordered protein MEG-3" for further consideration at *eLife*. Your revised article has been favorably evaluated by Vivek Malhotra (Senior Editor), and Julie Ahringer, (Reviewing Editor).

We are satisfied with your revisions except for your response to point 3, which needs to be addressed before acceptance, as outlined below:

The single sentence included in the Introduction is not sufficient to address point 3, requesting that you reconcile the findings in this paper with your study of MEG-3 in Wang et al. (*eLife*). Please explain (at least briefly) in the Discussion how the new results in this paper fit with your results in Wang et al. on MBK-2 and PPTR-1/2, for example, including your ideas above that MBK-2/PPTR-1/2 may act permissively. It will be confusing to readers that you have only in passing mentioned your previous results on MEG-3 in this paper.

---

## [Author Response]

*Essential revisions:*

*1) Quantification of granule number and size to back up qualitative claims. —The images in Figure 1 and Figure 1—figure supplement 1 on the influence of pgl mutants on Meg 3 granules appear quite different. In Figure 1 there seems to be a weakening of localization and granule size while the granules seem very similar to wildtype in Figure 1—figure supplement 1. Could the authors develop a more quantitative description for both sizes and numbers? Granule size seems also differently affected in mex 5/6 mutants for Pgl or Meg granules, again is this just the images presented or a consistent difference.*

The embryos shown in Figure 1 and previous Figure 1—figure supplement 1 (now Figure 1—figure supplement 1) are in different stages (1-cell versus 4-cell) and are representative. As noticed by the reviewer, we consistently see smaller/weaker MEG-3 granules in *pgl-1/3* zygotes, and this difference is most striking in the 1-cell stage and becomes less striking by the 4-cell stage. We previously reported this finding in Wang et al., 2014. The point of the experiment here, however, is that *pgl-1/3* are not required to localize the MEGs to the posterior in the 1-cell stage or to the P blastomere in the 4-cell stage, which is clearly illustrated Figure 1 and Figure 1—figure supplement 1.

*2) This careful study is in striking conflict to a study published by Hyman and colleagues jut last month in Cell. In this study a role for Pgl proteins akin to the function of Meg3/4 proposed here is populated. The authors briefly discuss this in the present manuscript, but a more complete discussion would be helpful. In particular, in vitro Pgl proteins can phase separate and recruit other granule components but apparently in vivo granule formation depends on Meg3/4. Can this be related to a functional significance for germ line development?*

We have now included a more complete discussion of the Hyman paper in the Discussion. The Hyman paper is a theoretical study that makes several predictions that are not consistent with in vivo observations. For example, the Hyman model predicts that the RNA binding domain of PGL-3 is essential for PGL-3 to form granules in the posterior cytoplasm in response to MEX-5. In fact, Hanazawa et al., showed in 2011 that PGL-3 lacking its RNA binding domain still forms posterior granules in vivo. The Hyman model also predicts that PGL-3 responds directly to the MEX-5 gradient. In fact, we find that PGL-3 asymmetry requires *meg-3/4* and does not correlate temporally with the MEX-5 gradient. Finally, the Hyman model assumes that PGL-3 is the driver of P granule asymmetry. In fact, neither PGL-3 (nor its paralog PGL-1) are required for P granule asymmetry: zygotes lacking PGL-1 and PGL-3 still assemble MEG granules in the posterior. Future genetic studies will be needed to compare the different contributions of MEG and PGL proteins to germline development, but with respect to P granule asymmetry in zygotes, the MEG proteins are the drivers.

*3) The findings in this paper are a significant departure from the conclusions this same group published two years ago in eLife showing that MEG-3 is a substrate of the kinase MBK-2 and the phosphatase PPTR-1/2 (Wang, et al.). In the Wang study, phosphorylation of MEG-3 by MBK-2 in the anterior of the C. elegans zygote promoted P-granule disassembly, while dephosphorylation by PPTR-1/2 promoted P-granule assembly. Attempts to reconcile these findings with the new discovery that asymmetric MEG-3 distribution and assembly now depends on an mRNA-binding competition with MEX-5 need to be included.*

The Wang et al. paper showed that assembly/disassembly of P granules requires the activities of MBK-2 and PPTR-1/2. That study, however, did not claim that these regulators function in a spatially-restricted manner. In fact, in the Discussion section of Wang et al., we discussed the possibility that MBK-2/PPTR-1/2 act only *permissively* to stimulate granule assembly/disassembly throughout the cytoplasm and that MEX-5 is responsible for the patterning. We now clarify this point in the Introduction.

*4) A central argument that distinguishes this paper from Saha et al. is that MEX-5 drives P-granule disassembly through MEG-3 instead of PGLs. Two findings support this. The first is that posterior enrichment of MEG-3::OLLAS is observed before posterior enrichment of PGL-1. PGL-3 is not examined and should be as the Saha et al. paper showed a specific PGL-3 response. There are available PGL-3 antibodies to test this. The second finding is that MEG-3 is still enriched in the posterior of pgl-1/3 RNAi embryos; however, the degree of knock down is not confirmed so PGL-1/3 may still be present. Co-stains of MEG-3/4 (anti-FLAG/OLLAS) with anti-PGL-1/3 in pgl-1/3(RNAi) embryos should be straight forward with existing antibodies.*

We have added the data requested by the reviewer. Using anti-PGL-3 antibodies, we show that, like PGL-1, PGL-3 does not localize properly in *meg-3/4* zygotes. Second, we clarify that the *pgl-1/3* zygotes shown in Figure 1 and Figure 1—figure supplement 1 are derived from *pgl-3(bn104)* hermaphrodites treated with *pgl-1(RNAi)* and we include the control staining with anti-PGL-1 to demonstrate loss of PGL-1 protein (Figure 1—figure supplement 1). These data demonstrate unequivocally that PGL-1 and PGL-3 are neither necessary nor sufficient for granule asymmetry.

*5) Findings presented in Figure 2 need clarification. It is not immediately clear that anterior MEX-6 is sufficient to prevent MEG-3 granules in the text. Nomenclature and labels in Figure 2 need improved to better sync with the text.*

We now include an additional control figure (Figure 2—figure supplement 1) that shows that MEX-6 is sufficient to localize MEG-3 in the absence of MEX-5. We also clarify this in the text. We have also made sure labels match between figures and text.

*6) It is not obvious as to why the ZF- mutated MEX-5 (R247E and K318E) also does not form a gradient, even though it has the necessary serine at position 404.*

Griffin et al., 2011 reported that the MEX-5 RNA binding domain is required for gradient formation. We have added an explanation of this to the text.

*7) Figure 1 – should specify for clarity what timepoint this is (before, during, or after polarization).*

Done.

[Editors' note: further revisions were requested prior to acceptance, as described below.]

*We are satisfied with your revisions except for your response to point 3, which needs to be addressed before acceptance, as outlined below:*

*The single sentence included in the Introduction is not sufficient to address point 3, requesting that you reconcile the findings in this paper with your study of MEG-3 in Wang et al. (eLife). Please explain (at least briefly) in the Discussion how the new results in this paper fit with your results in Wang et al. on MBK-2 and PPTR-1/2, for example, including your ideas above that MBK-2/PPTR-1/2 may act permissively. It will be confusing to readers that you have only in passing mentioned your previous results on MEG-3 in this paper.*

We now have incorporated a discussion of our earlier findings to explain how phosphorylation and RNA-induced phase separation cooperate to regulate P granule dynamics.

We have also expanded the last paragraph of the Discussion to outline remaining questions for future investigations.